# Icatibant Acts as a Balanced Ligand of MRGPRX2 in Human Skin Mast Cells

**DOI:** 10.3390/biom15091224

**Published:** 2025-08-25

**Authors:** Zhuoran Li, Jean Schneikert, Gürkan Bal, Torsten Zuberbier, Magda Babina

**Affiliations:** 1Institute of Allergology, Charité—Universitätsmedizin Berlin, Corporate Member of Freie Universität Berlin and Humboldt-Universität zu Berlin, Berlin, Germany; zhuoran.li@charite.de (Z.L.); jean.schneikert@charite.de (J.S.); guerkan.bal@charite.de (G.B.); torsten.zuberbier@charite.de (T.Z.); 2Fraunhofer Institute for Translational Medicine and Pharmacology ITMP, Immunology and Allergology, Berlin, Germany; 3Department of Dermatology, The Second Hospital of Tianjin Medical University, Tianjin, China

**Keywords:** mast cells, MRGPRX2, degranulation, icatibant, drug allergy, desensitization, signaling, LAD2 cell line

## Abstract

MRGPRX2 (Mas-related G protein-coupled receptor member X2) is implicated in mast cell (MC)-driven disorders due to its ability to bind diverse ligands, which may be G-protein-biased or balanced, with the latter activating both G-proteins and the β-arrestin pathway. Icatibant, a peptide drug, produces injection-site reactions in most patients and is used experimentally to probe MRGPRX2 function in skin tests. While reported to be G-protein-biased, it is unknown how skin MCs respond to icatibant, although these are the primary target cells during therapy. We therefore compared responses to icatibant with those induced by the balanced agonist substance P (SP) in skin MCs. Degranulation and desensitization were assessed via β-hexosaminidase release, receptor internalization by flow cytometry, and downstream signaling by immunoblotting. Skin MCs degranulated in response to SP and icatibant, relying on Gi proteins and calcium channels; Gq and PI3K (Phosphoinositide 3-kinase) contributed more strongly to exocytosis following icatibant, while JNK (c-Jun n-terminal kinase) was more relevant for SP. Both agonists activated ERK, PI3K/AKT, and (weakly) p38. Surprisingly, and in contrast to the LAD2 (Laboratory of Allergic Diseases 2 mast cell line) MC line, icatibant was at least as potent as SP in eliciting MRGPRX2 internalization and (cross-)desensitization in skin MCs. These findings suggest that icatibant functions differently in primary versus transformed MCs, acting as a fully balanced ligand in the former by triggering not only degranulation but also receptor internalization and desensitization. Therefore, not only the ligand but also the MRGPRX2-expressing cell plays a decisive role in whether a ligand is balanced or biased. These findings are relevant to our understanding of icatibant’s clinical effects on edema and itch.

## 1. Introduction

Mas-related G-protein-coupled receptor member X2 (MRGPRX2) has entered the focus of mast cell (MC) research as the chief receptor mediating clinical pseudo-allergic reactions, which are often caused by the direct activation of MCs but do not involve specific immunoglobulins produced by the adaptive immune system [1,2,3,4,5,6,7,8,9]. MRGPRX2 can bind a plethora of ligands, classified by distinct criteria. Classification is based on the source, i.e., endogenous (e.g., neuropeptides such as substance P, vasoactive intestinal peptide, or cortistatin) versus exogeneous (e.g., drugs such as ciprofloxacin, morphine, vancomycin, or atracurium), by chemical nature (small molecules, e.g., icatibant, compound 48/80 versus peptides, e.g., LL-37, β-defensins), or by the chemical groups in the ligands and amino acids within MRGPRX2 forming contacts, as well as by the signaling cascades elicited [10,11,12,13]. The latter distinction considers whether the respective agonist activates only G-proteins (G-protein-biased ligand) or also the β-arrestin pathway (balanced ligand).

G-proteins and β-arrestins are both key regulators of G-protein-coupled receptor (GPCR) signaling. When a ligand binds to a GPCR, the receptor undergoes a conformational change. This activates heterotrimeric G-proteins (Gα, Gβ, and Gγ) by facilitating guanosine diphosphate (GDP)–guanosine triphosphate (GTP) exchange on the Gα subunit. Activated G-proteins then trigger downstream signaling pathways. β-Arrestins are multifunctional proteins that regulate GPCR activity in two main ways: GPCR desensitization and internalization and G-protein-independent (β-arrestin-mediated) signaling. After prolonged activation, GPCRs are phosphorylated by G-protein-coupled receptor kinases (GRKs) at their intracellular domains. This phosphorylation recruits β-arrestins (β-arrestin-1 and β-arrestin-2), which sterically block further G-protein coupling, leading to desensitization (reduced receptor responsiveness). β-Arrestins also mediate receptor internalization via clathrin-mediated endocytosis, leading to receptor recycling or degradation [10,11]. Herein, the two β-arrestins (β-arrestin-1 and -2) can assume distinct functions. In skin MCs, β-arrestin-1 orchestrates internalization, while β-arrestin-2 is involved in signal termination but does not organize internalization [14,15]. Many known ligands are balanced, including neuropeptides like substance P (SP), opiates like codeine, and the best-investigated agonist compound 48/80, but the precise level of β-arrestin versus G-protein activation may still differ, even across “balanced” ligands [12,15,16].

An important and clinically relevant G-protein-biased ligand is icatibant, a bradykinin B2 receptor antagonist used to treat hereditary angioedema [17]. By activating MRGPRX2, it can provoke injection-site edema and itch as side effects during therapy [18] and in intradermal tests [19]. Its classification as biased is based on studies in cell lines such as the HEK293-derived TANGO Luciferase Assay cell line (HTLA) and the Laboratory of Allergic Diseases 2 mast cell line (LAD2), naturally expressing MRGPRX2 [20,21,22], or MRGPRX2 transfectants [20,23,24,25,26]. It is unknown how icatibant acts in physiologic MCs. Although not easily obtainable, skin MCs are among the most significant physiologic expressers of MRGPRX2 in humans [27,28,29,30].

Here, we elaborate on the stimulatory patterns, signaling, and counter-regulatory mechanisms of icatibant in physiologic and transformed MCs side by side. Unexpectedly, icatibant acts as a balanced ligand in skin MCs and elicits not only degranulation but also internalization and desensitization. Since the same MRGPRX2 ligand can operate differently in skin MCs versus cell lines, modeling of the drug’s mechanisms at edema and itch elicitation will require the use of physiologic MCs.

## 2. Materials and Methods

### 2.1. Cells and Treatments

Foreskins were obtained from circumcisions with the written informed consent of the patients or their legal guardians, as described [31,32,33]. The study was approved by the Ethics Committee of the Charité Universitätsmedizin Berlin and experiments were conducted according to the Declaration of Helsinki’s principles. MCs were isolated as described [34,35]. Briefly, foreskin samples (normally pooled from 2–15 donors) were cut into strips and treated with dispase at 3.5 U/mL (BD Biosciences, Heidelberg, Germany) at 4 °C overnight. The epidermis was removed; the dermis was then chopped to homogeneity and digested with an enzyme cocktail containing 1.5 mg/mL of collagenase type 1 (Worthington, Lakewood, NJ, USA), 0.75 mg/mL hyaluronidase type 1-S (Sigma, Steinheim, Germany), and DNase I at 10 µg/mL (Roche, Basel, Switzerland) at 37 °C in a shaking water bath for 75 min. The cells were separated from the remaining tissue by filtration and incubated with anti-human c-Kit microbeads for positive selection by Auto-MACS (both from Miltenyi Biotec, Bergisch Gladbach, Germany). MC purity (>98%) was assessed by toluidine blue staining (0.1% in 0.5 N HCl, Fisher Scientific, Berlin, Germany) and flow cytometry (see below) and is given in Appendix A. Viability was assessed (>99%) by trypan blue. Purified skin MCs were used ex vivo (within 18 h) in nearly all experiments, with no contact with cytokines to preserve their ex vivo profile (kept in Basal Iscove’s medium with 10% FCS, both from Biochrom, Berlin, Germany, overnight) [36]. In selected experiments (shown in Appendix A), skin MCs were used after long-term in vitro culture (4–5 weeks) in Basal Iscove’s medium with 10% FCS, non-essential amino acids (Sigma-Aldrich, St. Louis, MO, USA), 0.002% α-monothioglycerol (Sigma-Aldrich, St. Louis, MO, USA), SCF (100 ng/mL), and IL-4 (20 ng/mL) (both from Thermo Fisher Scientific, Waltham, MA, USA); SCF and IL-4 were freshly provided twice weekly when cultures were readjusted to 5 × 10^5^ /mL [37,38]. The LAD2 cell represents an MC sarcoma cell line; it was kindly provided by Dr. Metcalfe [39]. LAD2 cells were maintained in StemPro-34 medium (Thermo Fisher Scientific, Waltham, MA, USA) supplemented with L-glutamine, penicillin/streptomycin, and SCF (100 ng/mL), according to standard protocols [39]. If not otherwise specified, MCs were stimulated with saturating concentrations of icatibant (200 µg/mL, Shire Pharmaceuticals Ireland Limited, Dublin, Ireland) or SP at 60 µM (Bachem Holding, Bubendorf, Switzerland) in a suitable matrix, as described in the experimental procedures below.

### 2.2. β-Hexosaminidase Release Assay

Treatment with inhibitors and the detection of β-hexosaminidase were performed as described [32,39]. Cells were challenged as above for 60 min in PAG-CM buffer (PIPES-based buffer containing 0.003% human serum albumin (HSA) and 0.1% glucose) or kept without stimulus (control). Supernatants (SNs) were collected, and the pelleted MCs were rapidly frozen at −80 °C. After thawing, aliquots of 50 µL of 4-methyl umbelliferyl-n-acetyl-beta-D-glucosaminide (Sigma-Aldrich, Munich, Germany) solution at 5 µM in citrate buffer (pH 4.5) were mixed with the same volume of supernatant or lysate and incubated for 60 min at 37 °C. The reaction was stopped by 100 mM of sodium carbonate buffer (pH 10.7). The fluorescence intensity was measured at an emission wavelength of 460 nm after excitation at 355 nm. Percent β-hexosaminidase release = [fluorescence intensity SN/(fluorescence intensity SN + fluorescence intensity lysate)] × 100. Net release was calculated by subtracting spontaneous release.

### 2.3. Immunoblot Analysis

The detection of Akt, ERK1/2, and p38 was performed as described [35]. Ex vivo MCs and LAD2 cells were stimulated (at 5 × 10^5^ /mL in serum-free medium) as above. After centrifugation, cells were boiled in an SDS-PAGE sample buffer for 10 min. Lysates were separated through 4–12% Bis-Tris gels (Thermo Fisher Scientific, Berlin, Germany). After transferring to a blotting membrane and incubation with antibodies, proteins were visualized by a chemiluminescence assay (Weststar Ultra 2.0, Cyanagen, Bologna, Italy), and bands were recorded on a chemiluminescence imager (Fusion FX7 Spectra, Vilber Lourmat, Eberhardzell, Germany). The following primary antibodies, all purchased from Cell Signaling Technology (Frankfurt am Main, Germany), were used: anti-phosphorylated p38 (1:1000 dilution, Thr180/ Tyr182, #9211), anti-p38 (1:1000 dilution, #9212), anti-phosphorylated ERK1/2 (1:1000 dilution, T202/Y204, #9101), anti ERK1/2 (1:1000 dilution, #9102), anti-phosphorylated AKT (1:1000 dilution, Ser473, #9271), anti-AKT (1:1000 dilution, #4691), anti-α-actinin (1:1000 dilution,#6487), and cyclophilin B (1:25,000 dilution, #43603). Goat anti-rabbit IgG peroxidase-conjugated antibody was administered as the detection antibody (1:10,000 dilution, Merck, Darmstadt, Germany; #AP132P). The intensity of bands was quantified using ImageJ (version 2.0.0) (National Institutes of Health, Bethesda, MD, USA). The signal intensity of the proteins of interest was normalized to α-actinin and cyclophilin B, respectively, of the same membrane. To minimize the impact of different signal intensities across blots, which would have caused bias toward the ones with particularly strong signals, we calculated the mean of all lanes (i.e., all samples) of the membrane and determined the intensity of each normalized sample of the blot against this mean, as described [33]. The intensity of the protein of interest was finally defined as the mean value (±SEM) of all individual membranes (i.e., independent experiments), given as arbitrary units.

### 2.4. Flow Cytometry 

Flow cytometry was carried out as described [15]. Skin MCs and LAD2 cells were stimulated (at 5 × 10^5^ /mL in serum-free medium) as above for 1 or 24 h or kept without stimulus (control). MCs were blocked for 15 min at 4 °C with human AB serum (Biotest, Dreieich, Germany) and incubated with specific antibodies for 30 min at 4 °C, as described [32]. Anti-human MRGPRX2 (clone K125H4, BioLegend, San Diego, CA, USA) was used at 0.15 μg/mL; phycoerythrin-labeled mouse IgG2b (clone eBMG2b, eBioscience, San Diego, CA, USA) served as the isotype control. Surface expression was then determined on MACSQuant (Miltenyi Biotec, Bergisch Gladbach, Germany). Data were analyzed with the FlowJo analysis software v7.6 or v10.8 (FlowJo LLC, Ashland, OR, USA).

To assess skin MC purity, MCs were labeled with anti-KIT- (clone 104D2, Biolegend, San Diego, CA, USA) and anti-FcεRI-Abs (clone AER-37/CRA-1, Biolegend).

### 2.5. Statistical Analysis

Two-group comparisons were performed using unpaired or paired Student’s *t*-tests, or, for non-normally distributed data, the Mann–Whitney U test or Wilcoxon signed-rank test was applied. Repeated-measures one-way ANOVAs followed by Holm–Sidak’s or Tukey’s multiple-comparisons tests were used for comparisons involving three groups. EC_50_ values were calculated by non-linear regression analysis. Correlation analyses were performed using simple linear regression.

All statistical analyses were conducted using GraphPad Prism 8 (San Diego, CA, USA). A *p*-value less than 0.05 was considered statistically significant.

## 3. Results

### 3.1. Skin MCs and LAD2 Cells Degranulate with Icatibant

We investigated whether skin MCs were responsive to icatibant in comparison to LAD2 cells used as a positive control. Indeed, skin MCs degranulated with icatibant but with reduced efficiency compared to LAD2 cells (Figure 1a). A similar difference between the cells was found when SP was used as the stimulus (Figure 1b); this is in accordance with our recently described findings for balanced ligands [40]. Skin MCs and LAD2 cells robustly and comparably express MRGPRX2, as described [40]; an example of staining is also shown in Appendix A. After several weeks of culture, skin MCs responded even less strongly to both ligands compared to their ex vivo counterparts (Appendix A), which aligns with the reduced MRGPRX2 expression in pre-cultured skin MCs [36].

Dose–response curves in three independent skin MC preparations (each derived from a pool of donor skin samples collected on the same day) revealed EC50 values of 38–141 µg/mL. This indicates that responses to icatibant vary depending on the donors from whom the skin MCs were derived (Appendix A). Donor-dependent differences in the strength of MRGPRX2-dependent activation, even at an optimized (saturating) concentration, have been reported [15,31]. Responses induced by different MRGPRX2 ligands have also been documented to correlate across individual MC samples [15,31]. We tested whether this also applied to SP versus icatibant. Indeed, a strong correlation in degranulation efficiency was found between the stimuli for both skin MCs (Figure 1c) and LAD2 cells (Figure 1d).

### 3.2. Comparative Signal Transduction for SP and Icatibant

Having recently reported that extracellular signal-regulated kinases 1/2 (ERK1/2) and protein kinase B (AKT) are rapidly phosphorylated following MRGPRX2 activation by ligands like SP [14,33], we next studied signal transduction induced by icatibant vis-à-vis SP. Icatibant stimulated the same kinases but with slightly slower kinetics, while the signals tended to be more durable, since the blue curves were above the red ones at 60 min in skin MCs (Figure 2a). A similar tendency was found in LAD2 cells (Figure 2b), where signaling was enhanced for icatibant versus SP at the 30 min time point. Thus, icatibant induces signal transduction pathways comparably to SP yet with slightly altered kinetics.

Overall, signaling was more durable in LAD2 versus skin MCs, with no clear maximum, while 2 min typically marked the optimum in skin MCs, in accordance with previous findings [33,40,41,42]. We now show that this extends to a “biased” ligand and that both agonists behave similarly in this respect.

### 3.3. Shared and Distinct Signaling Prerequisites Underlie the Degranulation Responses Elicited by Icatibant and SP

We inquired as to whether the signaling modules responsible for degranulation differ between SP and icatibant. Gi (inhibitable by PTX, Pertussis toxin) and Ca++ channels (inhibitable by 2-APB, 2-aminoethoxydiphenyl borate) were of similar importance, with 85–90 or even nearly 100% of inhibition, respectively. La^3+^ (lanthanum ion)-inhibitable Ca++ channels played a minor role, and this was again comparable between stimuli. Interestingly, the Gq inhibitor YM-254890 and phosphoinositide 3-kinase inhibitor (PI3Ki) were more relevant when the stimulus was icatibant (Figure 3a). We therefore calculated the inhibition ratio for the Gq and the Gi inhibitor and found the increased significance of Gq over Gi in icatibant-triggered release (Figure 3b). Conversely, c-Jun n-terminal kinase (JNK) exhibited a stronger contribution when cells were stimulated by SP (Figure 3a). Therefore, there are overlaps but also some differences between MRGPRX2 ligands regarding the signals that underlie degranulation.

### 3.4. Icatibant Triggers MRGPRX2 Internalization in Skin MCs

In previous studies, icatibant did not activate β-arrestins in the TANGO (transcriptional activation following arrestin translocation) assay, which employs HEK293-derived TANGO Luciferase Assay cell line (HTLA) cells transfected with an MRGPRX2 construct, and β-arrestin independence was also found in MC lines [24]. We therefore explored whether MRGPRX2 was resistant to internalization by icatibant in physiologic MCs.

Surprisingly, icatibant triggered the disappearance of MRGPRX2 from the surfaces of skin MCs. In fact, the reduction upon icatibant was in a similar range to that elicited by SP after 1 h (Figure 4a). Conversely, a diminished reduction in MRGPRX2 cell surface expression was found in LAD2 cells, as expected, where the effect from SP was more pronounced (Figure 4b).

At 24 h post-stimulation, there was already recovery when SP was used as the stimulus, while icatibant led to a further reduction in skin MCs, additionally indicating differential kinetics of receptor downregulation between the two ligands (Figure 4a versus Figure 4c). Conversely, the pattern found in LAD2 cells was similar between 1 h and 24 h, with a more accentuated effect from SP remaining at the later point (Figure 4d). A direct comparison between the cell types can be found in Appendix A. Together, the data suggest that skin MCs are more sensitive to icatibant-induced receptor elimination, pointing to cell type-specific mechanisms governing MRGPRX2 turnover.

### 3.5. Icatibant Potently Desensitizes MRGPRX2 in Skin MCs

Since MRGPRX2 was internalized on icatibant stimulation, we asked whether this is accompanied by the desensitization and cross-desensitization of the receptor. Cells were pre-stimulated (or not) with SP or icatibant at time point 0 and then restimulated 24 h later to assess the degree of desensitization/cross-desensitization, as performed previously [15,40]. Interestingly, pre-stimulation with icatibant led to a diminished response to both agonists, reaching the same degree of desensitization as after SP, with no statistical difference between them (Figure 5a,b, blue versus red column).

The picture was highly distinct in LAD2 cells. Overall, desensitization was weaker in the cell line compared to skin MCs. Specifically, responses remained higher when icatibant was used as the first stimulus (blue columns in Figure 5c,d), whereby the difference between icatibant and SP was significant. We conclude that icatibant desensitizes MRGPRX2 less efficiently than SP in LAD2 cells, as would be expected for a biased ligand. Since desensitization by icatibant in skin MCs was as potent as that by SP, this further emphasizes the balanced nature of icatibant in physiologic MCs (Figure 4c,d).

## 4. Discussion

The discovery of MRGPRX2 marks a paradigm shift in MC biology, as it can explain how MCs are activated in a non-IgE-dependent manner, including in non-atopic subjects who suffer from MC-dependent hypersensitivity reactions all the same. In fact, many substances known for decades to activate MCs without awareness of the mechanism are now known to converge on MRGPRX2 [2,5,6,43,44]. The list of MRGPRX2 agonists is constantly growing, encompassing several hundred entities already [5,12,45]. A further level of regulation comes from the crosstalk between MRGPRX2 and other receptor networks, including KIT, the thymic stromal lymphopoietin receptor (TSLPR), and the receptor for interleukin-33 (IL-33R/ST2) [32,35,41,46,47,48,49]. Even though MRGPRX2 has low affinity, different agonists can support each other [50]. In addition, the acute co-stimulation of the IgE-dependent pathway results in additivity with the MRGPRX2-driven route, giving rise to multiple potential combinations in vivo [50].

Skin MCs express MRGPRX2 at highest level across the body [28,29,30,51] and are of pathophysiological significance in skin diseases and systemic reactions like anaphylaxis, in which skin symptoms are dominant [52]. To gain insights into how (presumably) biased ligands act on MRGPRX2 expressed within the skin MC membrane, we studied skin MC responses to icatibant, comparing it directly to SP. Surprisingly, icatibant did not behave like a biased ligand at all as it induced internalization and desensitization just like SP. SP, described as a balanced ligand in the literature [12], was reported to elicit MRGPRX2 internalization in skin MCs but less potently than codeine or compound 48/80 (two other balanced ligands) and with distinct kinetics [15,40]. This could be confirmed herein, since, at 24 h after SP treatment, the MRGPRX2 surface levels in skin MCs had almost recovered, while they were still at only ≈ 40% in LAD2 cells. Therefore, the pattern for SP versus icatibant was inverted between the cell types. Icatibant gave the largely expected pattern in LAD2 cells, as it was a less potent trigger of internalization and desensitization, behaving similarly to a biased ligand. Thus, the quality of a ligand to act as balanced or biased is not a definite feature but is context-dependent. MRGPRX2 may adopt distinct conformations in individual milieus of lipids and proteins, which are variably accessible and amenable to stabilization by different agonists.

This is further supported by the diverse behavior of SP (vis-à-vis icatibant) between the cells, especially in terms of MRGPRX2 reduction, which is substantially prolonged in LAD2 cells after SP stimulation despite LAD2’s overall lower tendency to internalize [40]. Although the MRGPRX2 levels recovered 24 h after SP stimulation in skin MCs, the cells were still desensitized. We therefore assume that numerical recovery does not mean full functionality and that a portion rather remains in a non-functional state after (re-)appearance at the surface, perhaps through the prolonged binding of β-arrestin-2 [14]. Considering our current and previous [40] data, a smooth transition or continuum across ligands is more probable than a categorical assignment to either the biased or balanced class. It is possible that each ligand leads to some degree of internalization and/or desensitization, albeit with different efficiencies (e.g., low for icatibant in LAD2 cells versus high for codeine in skin MCs, as shown in Ref. [40]).

We also asked whether the signaling events and molecular prerequisites underlying exocytosis differ between icatibant and SP. MRGPRX2 stimulation by SP, codeine, or compound 48/80 is known to trigger the rapid phosphorylation of ERK and AKT in skin MCs, while the induction of pp38 is weak due to its higher baseline phosphorylation [33,40,42]. This pattern was verified herein for SP and extended to icatibant. We found modest differences in the time courses between icatibant and SP, with events slightly delayed but longer-lasting with the drug. Despite these minor differences, the involvement of individual kinases in the degranulation process showed selected differences between agonists. JNK showed a greater contribution to SP-triggered degranulation, while PI3K (overall, the most meaningful downstream module [42]) was more significant when the stimulus was icatibant. The role of JNK in SP-elicited degranulation aligns with a previous report [48]. Moreover, YM-254890 suppressed icatibant-triggered degranulation more potently. This indicates that the bradykinin receptor antagonist preferentially stabilizes an MRGPRX2 conformation, which allows stronger coupling to Gq. This is further supported by the tendency of the Gi inhibitor PTX to be less inhibitory in the case of icatibant (although not significant); therefore, we also found an increased Gq/Gi inhibitory ratio for icatibant versus SP. The signaling prerequisites could, however, differ across MC subsets. We recently reported that ERK is strongly implicated in the SP-triggered degranulation of pre-cultured cutaneous MCs, while this necessity is much weaker in skin MCs ex vivo, where PI3K is dominant [42]. Similarly, the involvement of Gi versus Gq in MRGPRX2 signaling also depends on the cell type [53,54,55,56,57]. Here, we demonstrate that it is also influenced by the nature of the ligand in the same cell. Two recent structure–function studies reported different binding pockets being occupied by distinct ligands [13,58], and this is further supported by the differential amino acid requirements for the binding of distinct agonists (such as SP and cathelicidin/LL-37) [59]. In a recent study, molecular docking allowed the identification of three distinct patterns of small-molecule ligand–MRGPRX2 interactions based on the relevance of salt bridges/hydrogen bonding, π–cation, and π–π stacking between the chemical moieties of the ligand and amino acids of the receptor (GLU164, ASP184, PHE101, PHE170, TRP243, PHE244, and PHE257) [10].

We recently reported that LAD2 cells and skin MCs differ in their responses to MRGPRX2 ligands, with the cell line showing enhanced degranulation and prolonged signaling but poorer internalization and desensitization [40]. Since the MRGPRX2 expression levels are similar in LAD2 cells and skin MCs [40], as also shown in Appendix A, the overall reduced degranulation of skin MCs vis-à-vis LAD2 cannot be explained by reduced receptor expression. Instead, we recently showed that it is partially attributable to different equipment with β-arrestins [40]. 

The current study goes beyond this, however, as it reveals even greater complexity through the comparison between SP and icatibant. While a more active β-arrestin system can explain, at least in part, the greater propensity of skin MCs to internalize icatibant-bound MRGPRX2, it cannot explain why SP leads to more transient internalization in these cells compared to icatibant but is more potent and persistent in LAD2 cells. We therefore theorize that the conformations adopted by ligand-bound MRGPRX2 and the molecular events that ensue after the binding of the same ligand depend on the micromilieu in which MRGPRX2 is embedded, with its composition of proteins and lipids, and therefore on the MC type. Consequently, this also underscores a limitation in using LAD2 cells as a direct substitute for skin MCs in scientific or drug testing studies, as their differing signaling dynamics and measurable outcomes suggest that the results obtained from LAD2 cells may not fully capture the responses of primary MCs and will not even reveal whether a given ligand is of a balanced or biased nature. Since many approved medications activate MCs via MRGPRX2 as an off-target effect [5,12,45,60,61], it is probable that other drugs act like icatibant, with distinct reaction patterns found in LAD2 versus skin MCs. The bradykinin receptor antagonist can thus serve as a paradigm for a class of agonists that stabilize alternative MRGPRX2 conformations in physiologic compared to transformed cells. Assessing their mechanisms of action in relevant models is of substantial relevance considering that up to 18% of drug-like substances may activate MRGPRX2 as an off-target effect [45].

## 5. Conclusions

Our findings demonstrate that MRGPRX2 function strongly depends not only on the properties of individual ligands but also on the cellular context. Notably, icatibant—previously characterized as a G-protein-biased ligand—acts as a balanced agonist in primary human skin MCs, inducing not only degranulation but also receptor internalization and desensitization. In contrast, in LAD2 cells, icatibant behaves largely as a biased ligand. These findings underscore that the ability of MRGPRX2 ligands to function as biased or balanced agonists is not an intrinsic property but rather one that is dependent on the cellular environment. This highlights the critical importance of studying MRGPRX2 signaling in physiologically relevant human mast cells to more accurately elucidate how MRGPRX2-targeting drugs contribute to pseudo-allergic reactions and cutaneous symptoms.

## Figures and Tables

**Figure 1 biomolecules-15-01224-f001:**
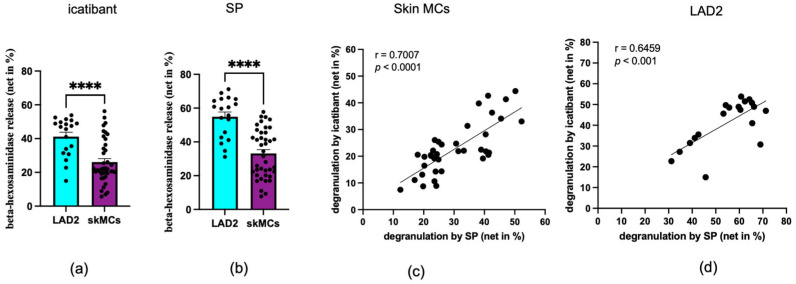
Skin MCs (purple) and LAD2 cells (cyan) degranulate in response to icatibant and SP. Skin MCs and LAD2 cells were stimulated with icatibant (**a**) or SP (**b**), and degranulation was measured by β-hexosaminidase release. Each dot represents an independent experiment (LAD2) or an independent skin MC preparation (donor pool). The data are presented as the mean ± SEM of *n* = 19–40. *****p* < 0.0001. Correlation analyses between icatibant- and SP-induced degranulation are shown for both skin MCs (**c**) and LAD2 cells (**d**), including linear regression lines, Pearson’s correlation coefficients (r), and corresponding *p*-values.

**Figure 2 biomolecules-15-01224-f002:**
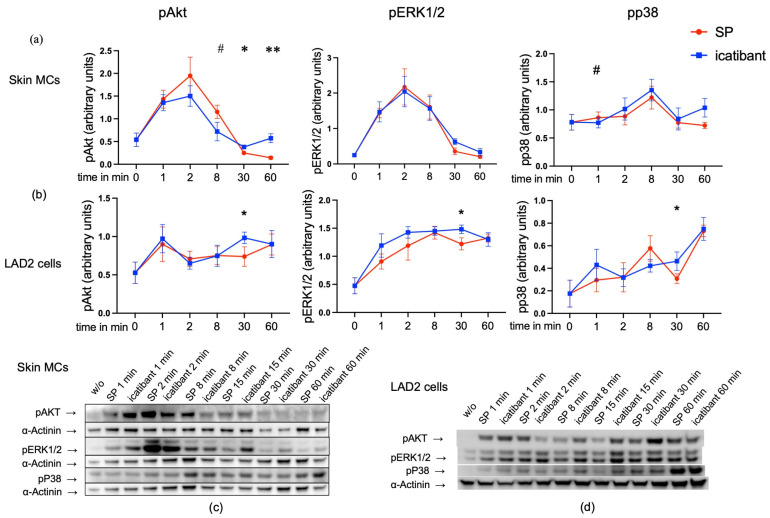
Phosphorylation kinetics are slightly shifted for icatibant versus SP. Phosphorylation of key kinases in skin MCs and LAD2 cells elicited by SP or icatibant after different times. (**a**,**b**) Band intensities of pAkt, pERK1/2, and pp38 were quantified and normalized as detailed in Materials and Methods. The data are the mean ± SEM (arbitrary units, calculated as explained in Materials and Methods) of 3–6 independent experiments, each using a different donor pool in the case of skin MCs. # SP > icatibant at *p* < 0.05; * icatibant > SP at *p* < 0.05, ** icatibant > SP at *p* < 0.01. p: phosphorylated. (**c**,**d**) Representative immunoblots (original images can be found in Appendix A).

**Figure 3 biomolecules-15-01224-f003:**
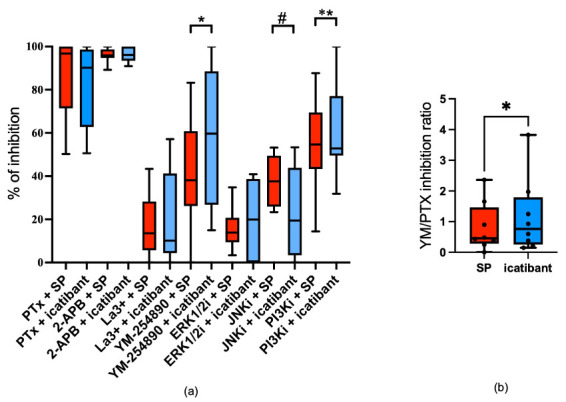
Signaling events contributing to degranulation by icatibant show shifts compared to SP. Skin MCs were pre-treated with 200 ng/mL PTX, 100 µM 2-APB, 1 µM La^3+^, 10 µM YM-254890, and inhibitors targeting ERK1/2 (SCH772984, 10 µM), JNK (SP600125, 5 µM), or PI3K (Pictilisib, 5 µM) and then stimulated by SP 60 µM (red) or icatibant 200 µg/mL (blue). β-Hexosaminidase release was determined as described in Materials and Methods. (**a**) Inhibition of degranulation by each inhibitor was compared between SP and icatibant stimulation. (**b**) The YM/PTX inhibition ratio was calculated for each MC preparation by dividing the % inhibition by YM-254890 by that of PTX (i.e., % inhibition by YM / % inhibition by PTX). The data are from 6–8 independent experiments, each using a different skin donor pool, and they are shown as box plots with whiskers, with the median given as a bar. ns: not significant, # SP > icatibant at *p* < 0.05; * icatibant > SP at *p* < 0.05, ** icatibant > SP at *p* < 0.01.

**Figure 4 biomolecules-15-01224-f004:**
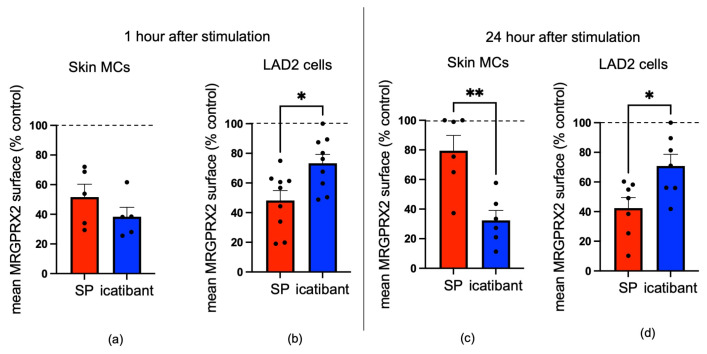
Icatibant and SP trigger MRGPRX2 Internalization in Skin MCs. (**a**,**c**) Skin MCs and (**b**,**d**) LAD2 cells were stimulated for (**a**,**b**) 1 h and (**c**,**d**) 24 h by SP or icatibant. Unstimulated cells served as controls. MRGPRX2 surface expression was measured and normalized to controls. Each dot represents an independent experiment (LAD2) or an independent skin MC preparation (donor pool). * *p* < 0.05, ** *p* < 0.01. Dashed line: no internalization (i.e., expression in the absence of an agonist).

**Figure 5 biomolecules-15-01224-f005:**
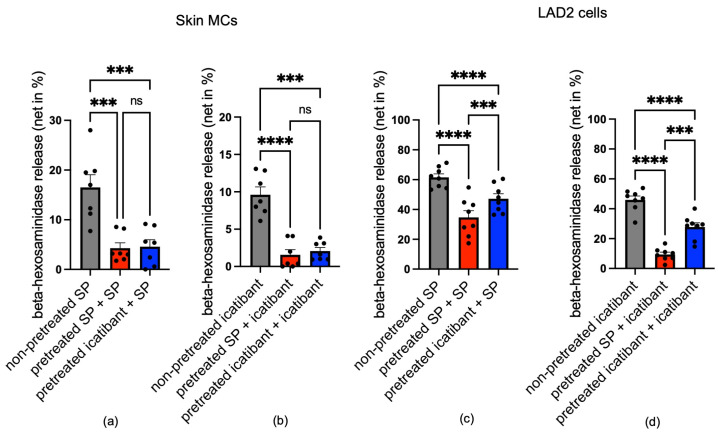
Icatibant and SP potently desensitize MRGPRX2 in skin MCs. (**a,b**) Skin MCs and (**c,d**) LAD2 cells were subjected to the specified pre-treatments at time point zero: SP (red) and icatibant (blue) versus no stimulus (non-pre-treated, grey). Cells were washed after 1 h and cultured in regular medium. After 24 h, cells were subjected to a second stimulation with SP or icatibant in a crossover setting. Net β-hexosaminidase release was assessed and normalized to the non-pre-treated group; mean ± SEM of 7–8 independent experiments. Each dot represents an independent experiment (LAD2) or an independent skin MC preparation (donor pool). ns: not significant, *** *p* < 0.001, **** *p* < 0.0001.

## Data Availability

The raw data supporting the conclusions of this article will be made available by the authors on request. No publicly archived datasets were consulted in connection with this study.

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
