# Peer review of "Icatibant Acts as a Balanced Ligand of MRGPRX2 in Human Skin Mast Cells"

_biomolecules, 2025, doi:10.3390/biom15091224_

Round 1
Reviewer 1 Report
Comments and Suggestions for Authors
The authors have presented an interesting study exploring how Icatibant acts as a balanced ligand of MRGPRX2 in human skin mast cells.
I have a few suggestions and comments that I believe could enhance the clarity and precision of the manuscript:
- While the authors mention the purity of the isolated cells, it is essential that they provide direct evidence to support this claim. I recommend including representative data (e.g., flow cytometry) showing cell purity and phenotypic characterization. This is particularly important given the use of ex vivo experiments, where post-isolation validation helps confirm that the observed effects can be attributed to the intended cell population.
- The unit "mg/ml" appears throughout the manuscript. For consistency with scientific conventions, please revise this to "mg/mL" (with a capital 'L' for liter).
- The phrasing in the sentence “Dose-response curves in three individual skin MC preparations (donor pools) revealed EC50 values of 38–141 µg/ml” is unclear and internally inconsistent. The term “donor pools” implies that each preparation was derived from multiple donors, which contradicts the use of “individual” in describing the MC preparations. If each sample corresponds to a single donor, the term “pool” should be removed. Conversely, if the preparations are pooled from multiple donors, the manuscript should clearly state this and adjust the description accordingly. Precision in terminology is essential for reproducibility and clarity.
- I recommend the authors reconsider using LAD2 as the sole positive control. To strengthen the study, key findings should be validated using additional primary cells (e.g., CD34⁺ peripheral blood cells or skin-derived mast cells from other healthy donors), and functional assays should include a known stimulus (e.g., anti-IgE or ionophore) applied to the same primary cells.
- Please note that multiple references, including reference number 34,57, and 58 appear in italics. This is inconsistent with the formatting of the rest of the reference list and should be corrected for uniformity.
Author Response
Comment 1: While the authors mention the purity of the isolated cells, it is essential that they provide direct evidence to support this claim. I recommend including representative data (e.g., flow cytometry) showing cell purity and phenotypic characterization. This is particularly important given the use of ex vivo experiments, where post-isolation validation helps confirm that the observed effects can be attributed to the intended cell population.
response 1: We thank the reviewer for this suggestion. Please note that we have been working with skin MCs for over 25 years and have extensively published their phenotypic, transcriptomic, proteomic and functional traits (as referenced in this paper). However, we agree such a figure is useful, so the reader does not need to consult previous publications if they are interested in verifying MC purity. We have therefore included a new supplementary Figure S1 showing toluidine blue staining to visually confirm the purity of the isolated MCs as well as flow-cytometric detection of KIT and FceRI expression in these cells, which are well-established markers for MCs.
The data demonstrate that the isolated cells are FcεRI⁺KIT⁺, confirming both the identity and high purity of the population used in the ex vivo experiments.
The changes in the Ms/figures are as follows:
*New Figure S1
Methods updated:
In 2.1. Cells and treatments
“MC purity (>98%) was assessed by toluidine blue staining (0.1% in 0.5 N HCl, Fisher Scientific, Berlin, Germany) and flow-cytometry (see below) and is given in supplementary Figure S1.”
In 2.4 Flow cytometry
“To assess skin MC purity, MCs were labeled with anti-KIT- (clone 104D2, Biolegend, San Diego, California; USA) and anti-FceRI-Abs (clone AER-37/CRA-1, Biolegend).”
Comment 2: The unit "mg/ml" appears throughout the manuscript. For consistency with scientific conventions, please revise this to "mg/mL" (with a capital 'L' for liter).
Response 2: We thank the reviewer for pointing this out. We have carefully revised the manuscript and corrected all instances of “mg/ml” to “mg/mL” to ensure consistency with scientific conventions.
Comment 3: The phrasing in the sentence “Dose-response curves in three individual skin MC preparations (donor pools) revealed EC50 values of 38–141 µg/ml” is unclear and internally inconsistent. The term “donor pools” implies that each preparation was derived from multiple donors, which contradicts the use of “individual” in describing the MC preparations. If each sample corresponds to a single donor, the term “pool” should be removed. Conversely, if the preparations are pooled from multiple donors, the manuscript should clearly state this and adjust the description accordingly. Precision in terminology is essential for reproducibility and clarity.
Response 3: We thank the reviewer for highlighting this issue. To clarify: each MC preparation used in the dose–response experiments (and all other parts of the manuscript) was indeed derived from a pool of human skin samples collected from multiple donors, with all samples in each pool being obtained on the same day. We use the term “pool” to reflect this composite nature of the preparations, which helps minimize inter-donor variability (though there is still some) and ensures sufficient cell numbers for downstream assays. Accordingly, we have revised the sentence in the Results section to read:
“Dose–response curves in three independent skin MC preparations (each derived from a pool of donor skin samples collected on the same day) revealed EC₅₀values of 38–141 µg/mL.”
This is also explained in the Methods (“MCs were isolated as described [34,35]. Briefly, foreskin samples (normally pooled from 2–15 donors) were cut into strips and treated with dispase at 3.5 U/ml (BD Biosciences, Heidelberg, Germany) at 4 °C overnight.”
Comment 4: I recommend the authors reconsider using LAD2 as the sole positive control. To strengthen the study, key findings should be validated using additional primary cells (e.g., CD34⁺ peripheral blood cells or skin-derived mast cells from other healthy donors), and functional assays should include a known stimulus (e.g., anti-IgE or ionophore) applied to the same primary cells.
Response 4: As for another cell type: while skin MCs are the gold standard in MRGPRX2 research, LAD2 cells are frequently used due to availability. So, this was the reason we compared the two. In addition, icatibant had been described as biased in cell lines, so it made sense to assess icatibant’s behavior in skin MCs in direct comparison with LAD2. Skin MCs are the cells that respond to icatibant in skin tests (skin prick or intradermal tests), and the side-effects from icatibant in patients during treatment (injection-site edema) also occur due on its action on skin MCs. It is therefore the system of primary interest.
However, since the reviewer also suggests skin-derived mast cells from other healthy donors, we realize that our descriptions have not been totally clear.
In fact, we have a myriad of donor pools in the study. Each dot in every figure corresponds to a distinct pool of donors. We are now pointing this out in all figure legends. (e.g., in Figure 1: “Skin MCs and LAD2 cells were stimulated with icatibant (a) or SP (b), and degranulation was measured by β-hexosaminidase release. Each dot represents an independent experiment (LAD2) or an independent skin MC preparation (donor pool). The data are presented as mean ± SEM of n = 19–40. ****P < 0.0001.”)
It would be utterly impossible to run the whole study with the same donor pool: There would not be enough MCs in one pool of even 15 or more foreskins. Besides, they would not be ex vivo anymore f the experiments were to be run over the course of months. In most instances the cells were just sufficient for onereadout, so each figure typically uses different donor pools.
Please note that a comparison between LAD2 and skin MCs makes also sense because the two systems express high levels of MRGPRX2 (newly added Figure S2) and respond to its ligation by various agonists. The difference is, however, that skin MCs are equipped with an effective beta-arrestin network to serve as a feedback mechanism; therefore, responses in skin MCs are typically weaker than in LAD2 cells. This was published recently here: PMID: 39481529. The figure is pasted below for convenience.
Figure from PMID: 39481529
While we appreciate the reviewer’s suggestion to include CD34⁺-derived MCs as an additional model, we need to point out that in our experience, CD34⁺-derived MCs typically express low (and highly donor-dependent) levels of MRGPRX2 and are thus poorly responsive to MRGPRX2 ligands. Therefore, they are not a good system to compare skin MCs or LAD2 cells to, since their MRGPRX2 levels are different. They will yield low responses, sometimes below the detection limit of the assays.
A better choice are culture-expanded skin MCs. These cells, however, likewise downregulate MRGPRX2 expression (and function) during culture, as published by us previously [PMID: 30091263], yet they still share many of the attributes of skin MCs, e.g., as published in PMID: 34329659, PMID: 35326404, PMID: 38201301.
We have now added a new supplementary figure S3 showing a) ex vivo versus precultured skin MC responses to SP and icatibant. As expected, cultured skin MCs responded less efficiently to both SP and icatibant.
Adaptations in the Ms:
Figure added as new Suppl. Fig. 3
In Methods: Joanna, copy the finalized part HERE
“Purified skin MCs were used ex vivo (within 18 h) in nearly all experiments with no contact to cytokines to preserve their ex vivo profile (kept in Basal Iscove’s medium with 10% FCS, both from Biochrom, Berlin, Germany, overnight) [36]. In selected experiments (shown in supplementary Figure S2), skin MCs were used after long-term in vitro culture (4-5 weeks) in Basal Iscove’s medium with 10% FCS, nonessential amino acids (Sig-ma-Aldrich, St. Louis, MO, USA), 0.002% a-monothioglycerol (Sigma-Aldrich, St. Louis, MO, USA), SCF (100 ng/mL), and IL-4 (20 ng/mL) both from (Thermo Fisher Scientific, Waltham, MA, USA); SCF and IL-4 were freshly provided twice weekly when cultures were re-adjusted to 5 × 105 /mL[37,38].”
In Results:
After several weeks of culture, skin MCs responded even less strongly to both ligands compared to their ex vivo counterparts (Supplementary Figure S3), which aligns with the reduced MRGPRX2 expression in precultured skin MCs [36].
As for other stimuli, we typically run different MRGPRX2 ligands and one non-MRGPRX2 stimulus but not everything in the same experiment for lack of cells (note that ex vivo skin MCs are available in low numbers only, hence also the pooling of foreskins across several donors). We have published on IgER-mediated versus MRGPRX2-mediated degranulation in several previous papers and the reader is referred to those references, e.g.,
Babina, M., et al., Allergic FcεRI- and pseudo-allergic MRGPRX2-triggered mast cell activation routes are independent and inversely regulated by SCF. Allergy, 2018. 73(1): p. 256-260.
Babina, M., et al., MRGPRX2 Is the Codeine Receptor of Human Skin Mast Cells: Desensitization through β-Arrestin and Lack of Correlation with the FcεRI Pathway. J Invest Dermatol, 2021. 141(5): p. 1286-1296.e4.
We have newly added this reference PMID: 30091263 and paste a figure from that paper herein
Babina M, Wang Z, Artuc M, Guhl S, Zuberbier T. MRGPRX2 is negatively targeted by SCF and IL-4 to diminish pseudo-allergic stimulation of skin mast cells in culture. Exp Dermatol. 2018 Nov;27(11):1298-1303. doi: 10.1111/exd.13762. Epub 2018 Sep 3. PMID: 30091263.
Note that crosslinking of the IgER is often a weaker stimulus in ex vivo skin MCs compared to MRGPRX2 ligands, while cultured equivalents show greater IgER- and lowered MRGPRX2 function. We paste the relevant figure here for convenience:
Fig from PMID: 30091263
Below we have additionally put together some of our most recent data as a courtesy to the reviewer. The datapoints given below are unpublished and are planned to be used in future publications. They show that the neuropeptides cortistatin-14 and SP yield very similar degranulation responses, while icatibant is also comparable (yet trending a bit lower). IgER-CL is typically weaker (as also apparent from the figure above / PMID: 30091263), while ionomycin is, again, similar to MRGPRX2 ligands.
Fig only for reviewer
These data confirm that skin MCs respond to both anti‑IgE and MRGPRX2‑receptor ligands under physiological conditions, albeit with different effectiveness.
Comment 5: Please note that multiple references, including reference number 34,57, and 58 appear in italics. This is inconsistent with the formatting of the rest of the reference list and should be corrected for uniformity.
Response 5: We thank the reviewer for pointing out this formatting inconsistency. We have carefully reviewed the reference list and corrected the formatting of references 34, 57, 58, and any other entries that were inadvertently italicized. All references now follow a consistent and uniform style, in accordance with the journal’s guidelines.
Reviewer 2 Report
Comments and Suggestions for Authors
Li et al. investigated the response of mast cells to icatibant. They used isolated skin mast cells and the LAD2 mast cell line, utilizing MRGPRX2 agonists, specifically the peptide drug icatibant and the neuropeptide substance P, as stimuli in vitro. Using a beta-hexosaminidase assay, they showed that icatibant triggers degranulation in both LAD2 and skin mast cells, with a relatively lower level seen in skin mast cells. Immunoblotting analyses of downstream signaling indicated that skin mast cells degranulated in response to substance P and icatibant, with the process dependent on Gi proteins and calcium channels. Following icatibant stimulation, the Gq protein and PI3K played a more significant role in exocytosis, whereas JNK was more pertinent for SP. Furthermore, unlike the LAD2 MC line, icatibant was at least as effective as SP in promoting MRGPRX2 internalization and (cross-)desensitization in skin MCs. Suggesting that icatibant exhibits distinct behavior in primary skin mast cells compared to the LAD2 cell line, functioning as a fully balanced ligand.
This study provides evidence of icatibant's effect on skin mast cell response. However, I have specific comments that I would like to see addressed.
a) Major comments:
- In the Materials and Methods section, please specify the cell culture medium utilized for LAD2 and skin mast cells. Is the same medium employed for stimulation to assess MRGPRX2 expression, immunoblotting analyses of downstream signaling, and the beta-hexosaminidase assay? Given that serum, cytokines, and growth factors can substantially affect mast cell responses, it is imperative to include this information.
- Figure 1 depicts the proportion of degranulation. It is crucial to assess the percentage of skin mast cells and LAD2 cells that display MRGPRX2 on their surfaces. The reduced response in skin mast cells might be attributed to a lower expression of MRGPRX2. Do skin mast cells with MRGPRX2 expression levels similar to those of LAD2 cells exhibit a comparable response?
- MRGPRX2 expression in skin mast cells versus LAD2 should be included in the discussion.
- What is the correlation between MRGPRX2 expression and the response to icatibant and substance P in skin mast cells and LAD2 cells, and how does the response to icatibant correlate to the response to SP?
- Figure 3 presents the signaling mechanisms that lead to degranulation triggered by icatibant and SP in skin mast cells. Is there a comparable signaling response in LAD2 cells when exposed to icatibant and SP?
b) Minor comments:
- Lines 126-127 “AKT(1:1000 dilution, Ser473, #9271), anti-AKT(1:1000 dilution, #4691), anti-α-actinin(1:1000 dilution,#6487), anti-β-actin(1:1000 dilution, #4967)” please include a space after “AKT”, “actinin” and “actin” before the brackets.
Author Response
Comments 1: In the Materials and Methods section, please specify the cell culture medium utilized for LAD2 and skin mast cells. Is the same medium employed for stimulation to assess MRGPRX2 expression, immunoblotting analyses of downstream signaling, and the beta-hexosaminidase assay? Given that serum, cytokines, and growth factors can substantially affect mast cell responses, it is imperative to include this information.
Response 1:
We thank the reviewer for this important comment. Please note that the skin MCs used in this paper were not cultured but used ex vivo (within 18 h after purification). They were only kept overnight in minimal medium (basal Iscove with supplements, but no SCF/IL-4) and stimulated directly on the next day. The absence of SCF/IL-4 is crucial to minimize culture-induced alterations, especially the known inhibitory effect of SCF/IL-4 on MRGPRX2 function in skin MCs [PMID: 28859248, PMID: 30091263].
We agree that preceding culture and the medium composition have an effect, as published by us previously [PMID: 30091263]. Following up on a suggestion from reviewer 1, We have now added a new supplementary figure 3 to demonstrate that the culture of skin MCs reduces responses to MRGPRX2 ligands (the order of degranulation efficiency is therefore LAD2 cells > ex vivo skin MCs > precultured skin MCs). However, all other findings of this paper (in particular all data shown in the previous, non-revised version) have used ex vivo skin MCs as the only system.
The media used for LAD2 cells and skin MCs (as mentioned, cultured skin MCs were only used for one supplementary Fig, that has been newly added to this revision) have been updated in the Methods section.
These details have been incorporated, as follows:
“Purified skin MCs were used ex vivo (within 18 h) in nearly all experiments with no contact to cytokines to preserve their ex vivo profile (kept in Basal Iscove’s medium with 10% FCS, both from Biochrom, Berlin, Germany, overnight) [36]. In selected experiments (shown in supplementary Figure S2), skin MCs were used after long-term in vitro culture (4-5 weeks) in Basal Iscove’s medium with 10% FCS, nonessential amino acids (Sigma-Aldrich, St. Louis, MO, USA), 0.002% α-monothioglycerol (Sigma-Aldrich, St. Louis, MO, USA), SCF (100 ng/mL), and IL-4 (20 ng/mL) both from (Thermo Fisher Scientific, Waltham, MA, USA); SCF and IL-4 were freshly provided twice weekly when cultures were re-adjusted to 5 × 105 /mL[37,38]. The LAD-2 cell represents a recently established MC sarcoma cell line that was kindly provided by Dr Metcalfe [39]. LAD2 cells were maintained in StemPro-34 medium (Schwerte, Germany) supplemented with L-glutamine, penicil-lin/streptomycin, and SCF (100 ng/mL), according to standard protocols [39]. If not otherwise specified, MCs were stimulated with saturating concentrations of icatibant (200 µg/mL, from Shire pharmaceuticals Ireland Limited), or SP at 60 µM (Bachem Holding) in a suitable matrix as described in the experimental procedures below.”
Importantly, all stimulations—including those for β-hexosaminidase release, immunoblotting, and flow cytometric analysis—were performed in either stimulation buffer or serum/cytokine-free media, not in the full culture medium. This is routinely done, also to avoid confounding effects of serum components, cytokines, or growth factors. The matrices used are given in the respective paragraphs.
Comments 2: Figure 1 depicts the proportion of degranulation. It is crucial to assess the percentage of skin mast cells and LAD2 cells that display MRGPRX2 on their surfaces. The reduced response in skin mast cells might be attributed to a lower expression of MRGPRX2. Do skin mast cells with MRGPRX2 expression levels similar to those of LAD2 cells exhibit a comparable response?
Response 2:
We thank the reviewer for this important comment. LAD2 and skMCs express similar levels of MRGPRX2 with 75-100% staining positive. In fact, we recently published a paper specifically dealing with differences between LAD2 and skMCs and the reason for the lower responsiveness of the latter despite similar expression levels (LAD2 cells have lower MRGPRX2 transcript levels, while cell surface expression seems similar) [PMID: 39481529]. We are pasting the published figure in this response letter:
The Fig. above is from PMID: 39481529
In the above paper [PMID: 39481529] we also revealed that beta-arrestins are at least one reason why the cells behave differently, with LAD2 cells showing globally greater degranulation responses but less internalization and desensitization.
Figure from PMID: 39481529
The above work [PMID: 39481529] only used balanced ligands, though. In the current manuscript, we included a so-called “biased” ligand for comparison with SP revealing that icatibant is not G-protein biased in skin MCs at all, while it still may be called biased in LAD2 cells. The important finding HERE is that also this trait (balanced versus biased) can change across cell types (irrespective of whether they respond more or less strongly to MRGPRX2 ligands overall).
Therefore, while we do verify our previous findings of reduced degranulation by skin MCs vs LAD2 in this study, it is not the major novel finding; the main messages of THIS paper are instead
-icatibant is a fully balanced MRGPRX2 ligand in skin MCs
-the ratios between SP/icatibant in terms of internalization vary between LAD2 and skMCs, i.e. SP leads to the strongest and most persistent internalization in LAD2 cells (also compared with other ligands published previously), while SP leads to less pronounced and transient internalization in skin MCs. With icatibant, we see the opposite, therefore, the internalization ratios between SP/Icat are not the same for the two cell types, and thus receptor turnover differs not only between ligands, but also across cells by which MRGPRX2 is expressed.
The latter cannot be explained by altered equipment with beta-arrestins alone.
If it were only the level of beta-arrestins that dictated the strength of internalization, we would expect that distinct ligands such as SP, icatibant, codeine all lead to overall stronger or poorer internalization in a given cell type compared to another cell type. The ratios between ligands, e.g., SP/icat would not change between cell types. But this is not what is observed. Therefore, on top of the distinct beta-arrestin levels there must be something else to dictate which of the ligands favor internalization in a cell-type dependent manner, i.e., why icatibant is better at internalization in skin MCs than SP (while the opposite is true in LAD2 cells).
We suspect that it is the way how MRGPRX2 is embedded in the cell membrane that leads to altered contacts between MRGPRX2 and its ligands in different MC types and hence altered downstream signaling. This is also part of the discussion section (“We therefore theorize that the conformations adopted by ligand-bound MRGPRX2 and molecular events that ensue after binding of the same ligand depend on the micromilieu in which MRGPRX2 is embedded with its composition of proteins and lipids and therefore on the MC type.”)
Taken together, it is not the altered MRGPRX2 expression that explains our findings, not even altered b-arrestin levels alone (for the reasons discussed above).
Changes in the Manuscript:
To show to the reader that MRGPRX2 is similarly expressed in skin MCs and LAD2 cells, we have added a new supplementary Figure (Figure S2)
In “Results”:
“Skin MCs and LAD2 cells robustly and comparably express MRGPRX2 as described [40]; an example staining is shown in supplementary Figure S2.”
Comments 3: MRGPRX2 expression in skin mast cells versus LAD2 should be included in the discussion.
Response 3: We have added a short paragraph to the revised Discussion section to clarify that the observed differences are not simply attributable to surface receptor levels but are more likely due to cell-type-specific signaling regulation downstream of MRGPRX2. It reads as follows:
“We recently reported that LAD2 cells and skin MCs differ in their responses to MRGPRX2 ligands, with the cell line showing enhanced degranulation, prolonged signaling but poorer internalization and desensitization [40]. Since MRGPRX2 expression levels are similar in LAD2 cells and skin MCs [40], as also shown in supplementary Figure S2, the overall reduced degranulation of skin MCs vis-à-vis LAD2 cannot be explained by reduced receptor expression. Instead, we recently showed it is partially attributable to a different equipment with β-arrestins [40].”
Comments 4: What is the correlation between MRGPRX2 expression and the response to icatibant and substance P in skin mast cells and LAD2 cells, and how does the response to icatibant correlate to the response to SP?
Response 4: This is a very good point. We know from previous studies that there is good (yet not absolute) correlation between MRGPRX2 expression levels and MRGPRX2 function within the same cell type, e.g., in skin MCs across donors. We paste one such figure below (unpublished).
This was confirmed in a publication from 2018, in which we showed that long-term culture of skin MCs in the presence of SCF and IL‑4 leads to downregulation of MRGPRX2 expression and simultaneously reduces degranulation responses to SP and other ligands [PMID: 30091263]. Thus, changes in receptor abundance translate into functional responsiveness in skin MCs. We paste the relevant part of the figure from PMID: 30091263 below.
Please also note a) that there is huge donor variability in the response to MRGPRX2 ligands (we believe this is also partially the reason why some individuals show strong pseudo-allergic reactions in the clinic, while others show weak or no reactions) and b) that responses induced by different ligands are highly correlated. In paper PMID: 28859248 we find this for SP versus c48/80 (Fig pasted below) and in paper PMID: 33058860 for codeine vs SP and codeine vs c48/80.
Figure from paper PMID: 28859248
Thanks to the excellent suggestion from the reviewer, we now also tested for correlation between icatibant and SP: Responses were highly correlated again, as would be expected for substances signaling via the same receptor.
The figure has been newly added to the Ms as part of Figure 1 and is also pasted below.
Changes in the Ms:
Incorporation of new Fig. 1c/d
+Update of Fig. 1a/b to include the same experiments as c+d (please note that several newer experiments have been added)
In Methods:
“Correlation analyses were performed using simple linear regression.”
In Results:
“Donor-dependent differences in the strength of MRGPRX2-dependent activation even at an optimized (saturating) concentration have been reported [15,31]. Responses induced by different MRGPRX2 ligands have also been documented to correlate across individual MC samples [15,31]. We tested whether this also applies to SP versus icatibant. Indeed, a strong correlation in degranulation efficiency was found between the stimuli for both skin MCs (Figure 1c) and LAD2 cells (Figure 1d).”
Comments 5: Figure 3 presents the signaling mechanisms that lead to degranulation triggered by icatibant and SP in skin mast cells. Is there a comparable signaling response in LAD2 cells when exposed to icatibant and SP?
Response 5: As shown in Figure 2, there is a comparable signaling response in both cells by quality, yet signaling is prolonged in LAD2 cells. Therefore, the curves have no clear maximum compared to skin MCs, for which we typically see a maximum at 2 min. A comparison between SP and icatibant shows modest trends towards longer signal duration with icatibant.
Regarding the signaling prerequisites that lead to degranulation (Fig. 3), our focus was on skin MCs, since these are the relevant responders in vivo; here, we also had data from previous papers to compare the requirements against [e.g., PMID: 33429916, PMID: 35326404]. Since these experiments are already quite extensive with one cell type, we did not typically use LAD2 cells simultaneously and concentrated entirely on skin MCs comparing different stimuli in the first place.
However, we have n=1 available also for LAD2 cells and the respective Fig. has been prepared to reveal the results to the reviewer.
In this experiment, PTX was not included. Besides, we can see the following:
*Ca++ channels were similarly relevant as in skin MCs with a nearly complete inhibition by 2-APB
*the Gq inhibitor YM-254890 had a slightly increased inhibitory potential against icatibant, which confirms the skin MC findings
*the JNK inhibitor had a (very) slight inhibitory effect for SP but not at all for icatibant (actually, even a slight increase was observed)
*what seems to be quite different compared to skin MCs: the PI3Ki was more potent against SP than against icatibant; in contrast, the ERKi was overall very potent in LAD2 cells compared to skin MCs, where effects were rather modest. In addition, icatibant was more strongly affected than SP. This is interesting, as we see a similar dichotomy between ex vivo skin MCs and precultured skin MCs, as published previously [PMID: 35326404]; the figure is pasted below and shows that the inhibitory ratio between PI3Ki and ERKi is shifted towards PI3K in the ex vivo variety, as repeated in the comparison between ex vivo MCs and LAD2 cells (albeit at n=1)
Please note that all signaling data of the current manuscript have been derived from ex vivo skin MCs.
Of course, from n=1 (LAD2) it is impossible to judge anything conclusive, since performance for 5-6 times, is required to assess similarities and differences, so we are not showing in the supplements. But we think the data are interesting and should be part of a future manuscript dealing with signaling component necessities across MRGPRX2 ligands and different types of MCs. We are considering this as a future plan also thanks to the reviewer’s hint.
Figures from publication PMID: 35326404
We have added the following to the Discussion section:
“Signaling prerequisites could however differ across MC subsets. We recently reported that ERK is strongly implicated in SP-triggered degranulation of precultured cutaneous MCs, while this necessity is much weaker in skin MCs ex vivo, where PI3K is dominant [42].”
Comments 6: Lines 126-127 “AKT(1:1000 dilution, Ser473, #9271), anti-AKT(1:1000 dilution, #4691), anti-α-actinin(1:1000 dilution,#6487), anti-β-actin(1:1000 dilution, #4967)” please include a space after “AKT”, “actinin” and “actin” before the brackets.
Response 9: We thank the reviewer for pointing out this formatting issue. We have added spaces after “AKT”, “actinin”, and “actin” before the corresponding brackets to ensure correct and consistent formatting in the revised manuscript.
Round 2
Reviewer 1 Report
Comments and Suggestions for Authors
Thank you for carefully addressing all of my comments and concerns. The revisions are thorough, and I have no further issues. I recommend acceptance in the current form.
Reviewer 2 Report
Comments and Suggestions for Authors
The authors have made all the requested changes and created a revised and more developed version of the manuscript. Thank you